# Investigating Industrial Effluent Impact on Municipal Wastewater Treatment Plant in Vaal, South Africa

**DOI:** 10.3390/ijerph17031096

**Published:** 2020-02-09

**Authors:** Eunice Iloms, Olusola O. Ololade, Henry J. O. Ogola, Ramganesh Selvarajan

**Affiliations:** 1Department of Environmental Science, University of South Africa—Florida Campus, Roodepoort 1709, South Africa; euniceiloms@yahoo.com (E.I.); henryogola@gmail.com (H.J.O.O.); 2Centre for Environmental Management, University of the Free State, Bloemfontein 9301, South Africa; shola.ololade@gmail.com; 3School of Agricultural and Food Sciences, Jaramogi Oginga Odinga University of Science and Technology, P.O. Box 210-40601, Bondo, Kenya

**Keywords:** industrial effluents, monitoring, wastewater treatment, water quality, Vaal River

## Abstract

Industrial effluents with high concentrations of toxic heavy metals are of great concern because of their persistence and non-degradability. However, poor operation and maintenance of wastewater treatment infrastructure is a great concern in South Africa. In this study, physico-chemical parameters and heavy metals (HMs) concentration of wastewater from five different industries, Leeuwkuil wastewater treatment plant (WWTP) inflow and effluent, and Vaal River water samples were monitored between January and September 2017, to investigate the correlation between heavy metal pollution and the location of industries and ascertain the effectiveness of the municipal WWTP. Physico-chemical variables such as pH, biological oxygen demand (BOD), dissolved oxygen (DO), chemical oxygen demand (COD), total dissolved solids (TDS) and electrical conductivity (EC) exhibited both temporal and spatial variations with the values significantly higher in the industrial samples. Inductively coupled plasma optical emission spectrometry (ICP-OES) results also showed that aluminium (Al), copper (Cu), lead (Pb) and zinc (Zn) were significantly higher in industrial effluents (*p* < 0.05), with only Zn and Al exhibiting significant seasonal variability. Statistical correlation analysis revealed a poor correlation between physicochemical parameters and the HMs compositional quality of wastewater. However, toxic HMs (Zn, Cu and Pb) concentrations in treated wastewater from WWTP were above the permissible limits. Although the WWTP was effective in maintaining most of the wastewater parameters within South African Green drop Standards, the higher Cu, Zn, Pb and COD in its final effluent is a concern in terms of Vaal river health and biological diversity. Therefore, we recommend continuous monitoring and maintenance of the WWTPs infrastructure in the study area.

## 1. Introduction

Industrialisation is key to the economic development and prosperity of any nation. However, the importance of industrialisation in the achievement of global and national economic growth, wealth and development notwithstanding, it can also be harmful to the environment. Rapid industrialisation is believed to be one of the main contributors to the pollution of environmental resources around the world [1], and South Africa is no exception [2,3]. While high-quality water is essential for most industrial processes, many of these processes generate large effluents of contaminated wastewater, whose safe disposal into receiving water bodies is of great environmental and health concern worldwide. For instance, competing demands on limited water resources coupled with awareness of the issues relating to water pollution, has led to considerable public debate about the environmental effects of industrial effluents discharged into the aquatic environments in South Africa [3,4]. 

Over the last two decades, there have been increased concerns on the potential adverse effects of industrial wastewater laden with high content of toxic heavy metal (HM) pollutants such as cadmium (Cd), nickel (Ni), lead (Pb), chromium (Cr), thallium (Tl) and mercury (Hg) [5,6]. More recently, contaminants of emerging concern (CECs) or micro-pollutants such as pharmaceuticals, personal care products (PCPs), pesticides, veterinary products, industrial compounds/by-products, food additives, and engineered nano-materials are posing a major threat to available water sources [7,8]. Among the pollutants, non-biogenic HMs such as Hg, Cd, Ar, Cr, Tl and Pb are highly hazardous and can have potentially toxic effects on living organisms [9]. These HMs are also generally persistent in wastewater treatment system because of their non-biodegradable and recalcitrant nature. Consequently, their presence in wastewater effluents and subsequent release to the environment may result in detrimental effect to both human health and aquatic ecosystems [2,10,11]. Therefore, continuous monitoring of wastewater and treatment plants before discharge into the environment is highly significant [12]. 

Industrial wastewater pollution is a notable problem in South Africa, a rapidly developing country where freshwater resources are short in supply. With just over 1200 m^3^/person/year of fresh water available for a population of around 50 million, the country is currently classified as water stressed [13]. Effluents generated from both industrial and domestic activities occupy the second position with respect to sources of water and presently constitute a major source of chemical and microbial pollution of South Africa’s water sources [3,11]. The Vaal Triangle is notably a major industrial hub of South Africa with diverse industries such as iron and steel, petroleum and coal oil companies, including gold mine industries. All these industries use hazardous and toxic chemicals in their industrial processes that can be considered as persistent and non-biodegradable. The Vaal River which passes through the Vaal Triangle is one of South Africa’s most important river, being the backbone of the country’s economy as it provides water to about 60% of South Africa, and 20 million people, representing approximately 45% of its population [4]. However, the water quality in the Vaal River as well as some of its tributaries is seriously affected because of adjacent mining, farming, urbanisation, and industrial activities. In the past two decades, it has become apparent that the Vaal River is increasingly under threat of pollution attributed to the aforementioned anthropogenic activities [14]. 

Several research studies indicate that most municipal WWTPs in South Africa rarely treat their wastewater to acceptable standards [15,16,17], while some engage in direct discharge of industrial effluents [18], thereby polluting receiving surface water sources. Furthermore, some of the WWTPs facilities are generally ill-equipped to remove large quantities of non-biodegradable waste and recalcitrant heavy metals that are eventually discharged onto the surface water sources [19]. However, there are no sufficient systematic studies that have been conducted to demonstrate a direct link and correlation between heavy metal pollution in water and the location of the industries within the Vaal catchment. Therefore, this study aims to: (1) investigate the correlation between heavy metal pollution in water and the location of industries in the Vaal triangle; and (2) ascertain the effectiveness of the Leeuwkuil WWTP in the treatment of industrial wastewater from the catchment before being discharge into the Vaal River.

## 2. Materials and Methods

### 2.1. Description of the Study Site

Vereeniging is a city located in the southern part of the Gauteng province in South Africa, very close to Vanderbijlpark (west), Three Rivers (east), Meyerton (north) and Sasolburg (south) as shown in Figure 1**.** Vereeniging is strategically located on the border of three provinces, consisting of Mpumalanga, Gauteng and Free State (Sedibeng District Municipality 2010). Leeuwkuil wastewater treatment works (WWTW) is located in Vereeniging within Emfuleni Local Municipality, which falls under Sedibeng District Municipality, and it offers effluent treatment services to the local communities. 

Built in 1958 to handle 83,050 PE (population equivalent), Leeuwkuil WWTP has a design capacity of 36,000 m^3^/day; with an installed 20,000 and 16,000 m^3^/day activated sludge plant and trickling filters, respectively [20]. In the plant, influent wastes undergoes: (1) pre-treatment (screening, grit, fats, oil and grease separation to remove large particles); (2) primary treatment to remove small particles (primary settling tanks, sedimentation, flotation); (3) secondary treatment (a multistage activated sludge process with integrated biological nutrient removal (BNR) for BOD, nitrates and phosphorous removal, trickling filters, anaerobic digestion); and (4) tertiary treatment (only chlorination done), before discharge into Vaal river (Figure 2).

Because of increased population and industrial development pressure in the area, the existing Leeuwkuil WWTW receives approximately 42,000 m^3^/d average daily waste flow (ADWF), which is 116% beyond the design capacity of the plant. By 2016, several of the biofilters were out of operation because of ageing infrastructure and poor maintenance. For example, the bio-filter plant with capacity of 16,000 m^3^/d were receiving on average approximately 2000 m^3^/d of the influence flow, with the excess hydraulic capacity of 40,000 m^3^/d being handled by activated sludge plant with built capacity of 20,000 m^3^/d [20]. In addition, extraneous flow (storm water and water ingress into the sewer system through leaking pipes or household fixtures) is estimated to be responsible for 40% of the current inflow into the plant. According to Sedibeng District Municipality [21], there are nine large water users (wet industries) ranging from battery industries, chemical manufacturing, ferrous and non-ferrous industries, tanking and car wash and food processing industries, and various small-to-medium industries discharging their effluent to Leeuwkuil WWTP, whose contribution to the hydraulic load on the WWTP is approximated at 10% (~4000 m^3^/d average daily waste flow).

### 2.2. Sample Collection

Triplicate water samples consisting of industrial effluents, treated water from WWTP, upstream and downstream water samples from Vaal River and potable water were collected from the eleven sampling points using sterile sampling bottles. The sampling points (upstream and downstream of the Leeuwkuil WWTP) are located along the Vaal River, while the inflow and final effluent were collected within the premises of Leeuwkuil WWTP as shown in Figure 1. For industrial effluents, samples were collected at the discharge points of each industry to the municipal sewer lines. These included: Industry 1 (Battery (lead acid) manufacturer); Industry 2 (Iron and steel—galvanize coating); Industry 3 (Iron and steel wire products); Industry 4 (Tanking and car wash); and Industry 5 (Steel wire manufacture). To understand the temporal effects, the samples were collected over a period of nine months between January and September 2017, representing the seasonal changes prevalent in South Africa (summer, autumn, winter, and spring). All the collected samples were preserved following APHA standards [22] and transported to the UNISA laboratory for analyses.

### 2.3. Determination of Physico-Chemical Parameters 

During sampling, water samples were analysed on-site in triplicate readings for temperature, pH, electrical conductivity, salinity, total dissolved solids (TDS), dissolved oxygen (DO) using calibrated multi-parameter probes (Hanna Instruments, version HI9828, SN 08334776, Woonsocket, RI, USA). The levels of chemical oxygen demand (COD) and biological oxygen demand (BOD_5_) were evaluated in the laboratory following standard methods as described by APHA (APHA 1995). Photometric measurement was used to measure COD using COD broad range kit (Hanna Instruments, HI839800, Woonsocket, RI, USA) performed according to the manufacturer’s protocol.

### 2.4. Heavy Metal Analysis

Elemental concentrations of collected wastewater and water samples were analysed using inductively coupled plasma optical emission spectrometry (ICP-OES, PerkinElmer Optima 5300 DV, Waltham, MA, USA) according to EPA Method 200.7 as described by Martin et al. [23]. Briefly, water samples were digested using 30 mL of HCl and 10 mL of HNO_3_ in a 3:1 ratio maintained at 100 °C until the samples were dissolved completely. The samples were then evaporated to a volume of 5–10 mL at room temperature after which they were transferred into 25 mL volumetric flasks containing 2.5 mL of Indium (internal standard) for metal analysis. All digested samples were filtered using 0.45 μm filter paper, prior to ICP-OES analysis. A calibration blank and an independent calibration and verification standards were analysed together with all samples to confirm the calibration status of the ICP-OES. The calibration curves with r^2^
** **>  0.999 were accepted for quantification resolution and the results were reported as the averages of triple measurements.

### 2.5. Statistical Analysis

Using statistical analysis software (SAS v 9.4, SAS Institute, Cary, NC, USA), analysis of variance (ANOVA) was used to compare the mean values of the tested parameters for all the different sampling sites, months and seasons at 0.05 level of significance using Tukey’s HSD as the post hoc test. The relationship between the physicochemical parameters and heavy metal constituents of the industrial wastewater and WWTP effluents was determined using two-tailed Pearson correlation analysis. Descriptive statistics was used to generate the means and standard deviation for the data sets.

## 3. Results and Discussion 

### 3.1. Industrial Wastewater Physicochemical Characteristics

The on-site physicochemical parameters recorded from different industrial and WWTP wastewater samples across all seasons sampled are presented in Table 1. The overall temperature ranged between 15.8–27.7 °C, the values being lower during winter (15–17 °C) than other seasons (20–29 °C). In summer, the highest sample temperature was recorded for Industry 3 compared to WWTP influent. A similar pattern was also noted during each of the other seasons (Table 1). However, no significant difference in temperatures was observed for the different sites (ANOVA, p = 0.879), consistent with the findings that temperature of wastewater generally mirrors the local prevailing ambient conditions [14,24,25]. According to Mahgoub et al. [26], a rise in the WWTP influent temperature above 35 °C is associated with negative effects on biological activity during the treatment process, that may reduce the waste treatment efficiency. Higher temperatures of the WWTP effluent is also undesirable as it can lead to the imbalance of aquatic organisms and its activities in the receiving water bodies.

Other physicochemical variables such as pH, BOD, DO, COD, TDS and EC exhibited both temporal and spatial variation. The pH values of Industry 2, WWTP influent and effluent discharges were in accordance with the acceptable limit set by Department of Water Affairs and Forestry [27], while, Industry 3, 4 and 5 had significantly higher pH values >9.5 (ANOVA, *p* < 0.0001), indicative of highly alkaline wastewater; the only exception was Industry 1 (battery industry), which was acidic (pH 4.6 ± 0.8). Consistent with our results, Howell et al. [28] reported that variations in wastewater pH is dependent on the main industrial activity and seasonality. On the other hand, a trend was observed albeit with significant differences for EC, TDS and salinity in the order Industry 5 > Industry 2 > Industry 4 > Industry 1 > industry 3 (*p* < 0.010, *p* < 0.0001 and *p* < 0.0001 for EC, TDS and salinity, respectively). Different industrial processes use an array of organic and inorganic compounds and release diverse end products that can influence effluents pH, EC, TDS and salinity. For instance, battery industry is associated with discharges of waste high in sulphuric acid leading to low pH (up to 1.6), heavy metals like lead and different inorganic salts that may elevate the TDS, EC and salinity contents [29].

Generally, carwash effluent (Industry 4) contains a number of contaminants such as sand/dust, salt, oils, grease, organic matter and metal ions that may also influence the resultant wastewater pH, TDS, EC, salinity and heavy metal composition [30,31]. Similar to our finding, other studies have reported that significantly higher EC, TDS and salinity occur during months of low precipitation (such as winter in South Africa), as increased evaporation leads to a more concentrated effluent [32,33,34]. Conversely, the dilution effect on the wastewater milieu during high precipitation months (summer) influences the resultant effluent composition (Appendix A). Higher values of EC and TDS in wastewater may be attributed to the presence of contaminants such as Na^+^, Ca^2+^, Mg^2+^, K^+^, Cl^−^, SO_4_^2−^ and HCO_3_^−^ and other metal salts [35]. Despite the final WWTP effluent, EC and TDS levels being lower than other sampling sites across all the seasons, the recorded levels did not meet the South Africa Green drop standard and permissible limits [27]. Therefore, our results imply that the Leeuwkuil WWTP is not effective in abstracting contaminants high in the aforementioned soluble salts, thus contributing to the observed high EC and TDS beyond the legal limits for final WWTP effluent.

In this study, values of DO in the industrial, WWTP inflow and effluents were observed to be between 1.0–2.7 mg/L, but varied significantly across the sampling points and seasons. Comparatively, the final WWTP effluent had significantly higher DO compared with other wastewater streams (ANOVA, *p* < 0.0056). The reported levels were still lower than the required standard of 8–10 mg/L before discharge into surface water [17,19]. Consistent with our studies, Momba et al. [16] and Edokpayi [37] recorded DO levels in the range of 3.26–4.57 mg/L in their investigation of the impact of inadequately treated effluents of four wastewater treatment facilities in Buffalo City and Nkonkobe Municipality of Eastern Cape and Limpopo Province, respectively, of South Africa. Similar to DO, the mean COD levels of the industrial effluents and WWTP inflow were significantly higher (ANOVA, *p* < 0.0001) than WWTP outflow. The South African guideline value for COD in wastewater is 75 mg/L, which exceeded in some of the sampling months in the WWTPs (Table 1). 

It is estimated that the Leeuwkuil WWTP handles 42,000 m^3^/d average daily waste flow that is 116% beyond the design capacity of 36,000 m^3^/d [20]. In addition, reported constant breakdown of the trickling filter plants because of ageing infrastructure, imply that the bulk of wastewater (approximately 40,000 m^3^/d average daily waste flow) are processed through activated sludge plant (ASP) having design capacity 20,000 m^3^/d average daily waste flow. The operation efficiency of an ASP is mainly dependent on organic loading rates (in terms of BOD) and hydraulic loading, which affects the dissolved oxygen (DO) demand in the aeration tank, biosolids wasting rate, optimal return activated sludge (RAS) rate, and solids settling and compaction characteristics [38]. In general, organic loading rates of 0.5–1.5 kg BOD/m^3^/day with hydraulic retention times of 5–14 h depending on the nature of the wastewater, is capable of giving BOD reductions of 90–95% in a conventional activated sludge process [39]. Based on the maximum hydraulic load of 40,000 m^3^/d average daily waste flow, the organic loading of the influent is 132 ± 4.2 kg BOD/m^3^/day. However, the calculated organic loading was not statistically different (*p* < 0.05) from that of WWTP effluent. This illustrates the inefficiency in treating wastewaters probably related to delivery of pollution loads that exceeds the treatment capacity of a municipal WWTP. 

The BOD: COD ratios can also be used for the characterization of wastewaters. Typically, municipal wastewaters have BOD: COD ratios between 0.2:1 and 0.5:1, with ratio being fairly steady for domestic sewage [38]. However, discharge of industrial effluents of variable composition and loading may fluctuate the ratio considerably. In this study, the calculated average BOD: COD ratio for WWTP influent was 0.02:1. Such very low BOD: COD ratios indicate high concentrations of nonbiodegradable organic matter in the influent that may be inefficiently degraded by the biological effluent treatment processes [25,38]. Overall, the aforementioned factors, process limitations and challenges within the Leeuwkuil WWTP account for the inefficiency to treat the wastewaters to acceptable South African Green Drop standards [36]. The organic load of wastewater and the treated effluent contributes significantly to oxygen demand level of the receiving water in terms of DO, COD and BOD. High organic load in untreated wastewater is associated with high BOD and COD leading to increased depletion of dissolved oxygen (DO). DO level below 5 mg/L would adversely affect aquatic ecosystem in the receiving surface water, and, therefore, the reported DO, COD and BOD levels of the final effluents is an environmental concern that needs addressing.

### 3.2. Heavy Metal Composition of Industrial Wastewaters

In this study, samples collected from five industrial effluents and Leeuwkuil WWTP (Inflow and final effluent) were analysed for the identification and determination of heavy metal concentrations. In total, 24 elements were detected at different concentrations (Appendix A). The results for Cd, Li, Mo, Te, Sb and V were not shown because of their very low detection in the samples. The total concentration of major heavy metals that are commonly found in the industrial wastewater, WWTP inflow and effluents samples and their corresponding discharge standards are presented in Table 2. 

Wastewater from Industry 1 had significantly higher levels of Al, Cu and Pb, followed by Industry 4, 5, 3 and 2, which could be ascribed to the main processes related to these industries. Physicochemical and elemental characterisation of the automotive battery industry effluent indicates that it is complex and strongly acidic in nature containing a variety of heavy metals above the legislated limits for discharge [40,41]. Heavy metal contents of Industry 1 were of the magnitude in the order of Al > Zn > Pb> Cu > Ni > Cr, with all metal content being consistently above legal permissible limits. However, significant season variability was only observed for Zn (ANOVA, *p* < 0.0001) and Al content (ANOVA, *p* = 0.042) (Figure 2). Consistent with our results, Vu et al. [40] reported an average concentration of Pb in battery wastewater of about 3–15 mg/L. Similarly, Ribeiro et al. [41], also reported higher concentration values for Fe (344 ± 96 mg/L), Zn (60 ± 17 mg/L) and Pb (22 ± 15 mg/L) and other heavy metals (Cr, Cu and Ni < 2 mg/L) in lead acid battery industry wastewaters. In contrast, metal industry effluents (Industry 2, 3 and 5) exhibited varying levels of heavy metals contamination above the permissible legal limits, with Industry 2 (galvanized iron coating industry) having characteristic higher Zn (56.7 ± 19.03 mg/L; ANOVA, *p* < 0.0001) and Cr content (0.80 ± 0.18 mg/mL). Ferrous and non-ferrous metal industry often undertake process such as acid pickling of metal surfaces to remove the oxide layer, rust, encrustations, inorganic contaminants or other impurities from ferrous metals, copper, or precious metals, resulting in wastewater high in heavy metals such as Cu, Ni, Zn, Cr among others [6]. On the other hand, Industry 4 (tanking/car wash) wastewater had an appreciable high content of Al, Zn and Ni compared to other industrial effluents. Several studies have also reported that carwash effluents are rich in pollutants of concern such as heavy metals [42,43]. Tekere et al. [30] also reported higher average Cu and Zn contents in carwash effluents in Gauteng Province of South Africa, mainly derived from brake pads and tyres. Because of their high polluting nature, wastewater from carwash should be pre-treated before disposal into municipal sewers. However, Tekere et al. [30] revealed that different carwash differ in the efficiency of their pre-treatment systems in terms of pollutant removal from effluents and the quality of their effluents significantly differs in the concentrations of most key parameters. Comparatively, the values reported in the study were much higher in terms of Cu content, but lower Zn level was reported in this study. Findings in this research seem to concur with the observations that carwash wastewater is predominantly characterised by heavy metal pollutants such as Zn, Cu in addition to other compounds of environmental concerns. 

In general, heavy metal composition of WWTP inflow and final effluents were comparatively lower than those of industrial effluents across all seasons (Figure 3). In Leeuwkuil WWTP, different industrial wastewater streams from the industrial catchment area are mixed with domestic sewage discharges and storm-water runoff before it reaches the treatment system. This practice may have a diluting effect, accounting for the low heavy metal composition of the influent wastewater observed in the study. However, a distinct reduction of the heavy metal concentrations between influent and effluent streams during a treatment process should be a good indicator of the WWTP efficiency in their removal. Unfortunately, results showed that there were no significant differences between the heavy metal content of the influent and effluent samples (Table 2).

Industrial water demand and wastewater production are sector-specific. Thus, the concentration and composition of the waste flows (mass/time) can vary significantly depending on the industrial process. In Leeuwkuil WWTP, it is estimated that industrial wastewater contribute to approximately 10% of influent hydraulic load [21]. Though not significant in capacity, such industrial wastewater may have high loads of a wide variety of microcontaminants such as heavy metals and non-biodegradable organics that may add to the complexity of wastewater treatment. Comparatively, heavy metals are not easily removed in a standard WWTP configuration, and often requires an additional tertiary treatment such as chemical precipitation, oxidation or coagulation techniques for their removal [6]. Similar to other urban WWTPs designed to cope mainly with the content of organic matter and nutrients typical of domestic effluents, Leeuwkuil plant has only chlorination (disinfection) step as only tertiary treatment before effluent are discharged into the Vaal River (Figure 2). Because of their recalcitrant nature, industrial effluent contaminants can affect the performance of the plants by hindering their biological functioning, particularly in cases where the WWTP was not designed to accept particular types of industrial effluent and/or where the characteristics of industrial effluents have changed over time [18]. Furthermore, toxicity, a higher nutrient content and a higher concentration of organic matter can induce changes in the balance of bacteria in various steps of the treatment leading to inefficiency of the biological treatment process [39]. 

A common issue also faced by Leeuwkuil and other urban WWTPs in South Africa relates to capacity overloads due to extraneous flow (storm water and water ingress into the sewer system through leaking pipes or household fixtures, broken pump stations due to ageing infrastructure), which has been reported to account up to 40% hydraulic load [19,20]. In addition to increasing the hydraulic load, such indirect effluents entering the WWTPs during storms initially increase pollutant concentrations that are then diluted. Some pollutants that are likely to increase in concentration during storm events include heavy metals, nitrogen and phosphorus which may affect the WWTP efficiency. The final treated wastewater from Leeuwkuil WWTP had heavy metal content (Zn, Cu and Pb) above the permissible limits [27,36]. These effluents are finally discharged into the Vaal River, and thus constitute a major point source of pollution to the river ecosystem. Hence the findings from this study indicate the limitation and/or inefficiency of Leeuwkuil WWTP in treating industrial wastewater high in both organic and inorganic loads, despite improving the DO, COD, BOD and heavy metal levels within the final effluent vis-à-vis the untreated industrial wastewater. Furthermore, in agreement with our findings, industrial activities were reported to influence the concentrations of heavy metals such as Pb, Cu, Mn, Zn, Fe and Cd above permissible limits by WHO, USEPA, EUC and EPA in rivers impacted by industrial wastewater in Nigeria [44], Ethiopia [45] and India [34]. 

The analysis of the interrelationship between physical parameters and heavy metals also gave an insight about the influence of the industrial effluents on the wastewater treatment plant. The results of bivariate two-tailed Pearson correlation analysis are presented in Table 3. Overall, there was a poor correlation between physicochemical parameters and the heavy metal compositional quality of wastewater. However, subtle observations could be discerned from the results. Among the physicochemical parameters, only pH had a moderate negative correlation with Cu (r = −0.61, *p* < 0.01) and Pb (r = −0.57, *p* < 0.01). Other significant moderate negative correlation observed between the effluent’s parameters included: BOD and DO (r = −0.47, *p* < 0.05); and salinity and TDS (r = −0.60, *p* < 0.01). Among the heavy metals, a significant positive correlation Pb (r = 0.96, *p* < 0.001) with Cu could indicate the same or similar source input. Consistent with our results, Zinabu et al. [45] and Ebrahimi et al. [33] reported positive correlation between heavy metal in rivers waters polluted with industrial wastewaters. Poor correlation between physicochemical parameters and the heavy metal content of WWTP influent and effluent wastewater was also noted (Table 4). Strong negative correlation was only observed for DO and COD (r = −0.95, *p* < 0.001), DO and TDS (r = −0.87, *p* < 0.01), DO and salinity (r = −0.90, *p* < 0.001), COD and salinity (r = −0.95, *p* < 0.001), and Al and Cu (r = −0.88, *p* < 0.01), whereas TDS and salinity, TDS and EC, and COD and TDS had moderate to strong significant correlation at *p* < 0.05.

### 3.3. Impact of Industrial Wastewater and WWTP Effluents on Vaal River 

To determine the changes in the Vaal River due to the discharge of wastewater, physicochemical parameters and heavy metal composition were investigated upstream and downstream of WWTP during the sampling period. These values were compared with background levels in potable water collected in two locations within the study (Table 5). The pH values of the Vaal River remained constant between 7.7–9.2 across all seasons, and did not show significant difference with potable water samples (ANOVA, *p* = 0.3521). As a consequence of different seasons, water temperature showed temporal differences, but was not significantly different between sampling sites. At the same time river water samples had significantly elevated levels of BOD, COD, TDS, EC, salinity, Al, Cu and Pb vis-à-vis background values in potable water (Table 5). Specifically, downstream river samples had significant higher levels of COD, TDS and Pb contents compared to upstream samples (*p* = 0.05), that are above the permissible levels by SANS 241-1 2015 standards [46]. There was ~two-fold increase in the level of mean COD (88 ± 57.2 mg/L vs. 162 ± 54.6 mg/L) and Pb content (0.098 ± 0.033 mg/L vs. 0.183 ± 0.041mg/L) in upstream and downstream samples, respectively. Interestingly, the background levels of Cu and Pb in potable water from the study site were below the detectable limits of the ICP-EOS, indicating WWTP effluent as the source of elevated Pb downstream. Our result is consistent with the findings of Osman and Kloas [47] who observed Pb concentrations to be higher in Nile River as a result of urban effluent draining into the river. On the other hand, the values of Zn, Cu and Al, though higher in downstream samples in contrast to upstream samples, were not significantly different. This suggests that the sewage treatment plant is not a major source of these trace metals. The Cu and Zn levels fell within the SANS [46] guidelines for potable water and irrigation water, which are between 0.2 and 5 mg/L, respectively. The EC values for downstream and upstream and potable water samples ranged from 368 to 763 μS/cm (Table 5). These values are higher compared to those reported by Jordan and Bezuidenhout [48] for the Vaal River in 2012. This implies that the rate of pollution has increased over the years possibly because of increased anthropogenic influences. Du Plessis [49] recently reported that there is increasing evidence that industrial effluents is associated with an increase in BOD, total dissolved solid and total suspended solid levels, toxic metals such as cadmium, chromium, nickel and lead, and faecal coliforms in the Vaal Barrage. Chigor et al. [50], also reported higher EC values than the current study which they attributed to the influence of pollution from land use activities within the Buffalo River Catchment in the Eastern Cape Province of South Africa. 

Overall, the heavy metal levels reported in this study were very low compared with those reported in polluted rivers linked to industrial and WWTP effluents [3,32,33,34,45,51]. One limitation of the study is that heavy metal composition was only done on water samples that generally gives lower heavy metal composition, particularly at higher pH condition. At higher pH, most heavy metals are removed from solution by precipitation process and get adsorbed onto the bottom sediment because of the production of hydroxide [52]. Although there was no significant difference in the pH of upstream and downstream samples, higher level of heavy metals downstream is a pointer that the concentration of the heavy metal may be higher in river sediments downstream, giving a better indication of the impact of the WWTP on the pollution of the river. Heavy metal pore-analysis of the river sediment would have helped to better define the ecotoxicological risk of the river [53].

Results from this study show that industrial wastewater and WWTP effluent are evidently burdened with toxic heavy metals like Pb that the municipal WWTPs are unable to treat efficiently, therefore constituting as pollutant in the Vaal River. Disposal of inadequately treated effluents from an array of industries is leading to the degradation in the quality of the Vaal River ecosystem, making the water unsuitable for drinking purposes, irrigation and aquatic life [2,54,55,56]. Hence, better control of the industrial sources is needed to help minimize the quantity of contaminated wastewater into already overburdened or inefficient municipal WWTPs. Currently, all industries are required by law to have pre-treatment system for their wastewater, accompanied with proper monitoring of the industries for compliance. Adherence to this law by industries would reduce the burden of treatment being placed on WWTPs. The improvement in the operations and maintenance of municipal wastewater and upgrading sewage treatment infrastructure, taking into account the complexity and recalcitrant nature of the emerging pollutants from industrial wastewaters, is urgently needed in South Africa to efficiently deal with the ever increasing demand of industrial development and associated wastewater. Finally, there is also a need to establish total maximum daily loads (TMDLs) for the common pollutants originating from the WWTPs to the Vaal river. According to EPA, TMDLs are critical to determine wastewater pollutant reduction target (maximum amount of a pollutant allowed into waterbody) and allocation of load reductions for WWTPs, that will ensure the receiving waterbody can meet water quality standards for that particular pollutant [57,58]. In South Africa, only annual TMDLs for phosphorus loadings to certain South African dams have been provisionally estimated [59,60], indicating the need for TMDLs to control the discharge of pollutants by the various point sources such as WWTPs. 

## 4. Conclusions

In this study, it was evident that there is a correlation between the effluent from industries within the Vaal River Catchment and the outflow from the Leeuwkuil WWTP, an indication that the industrial effluent significantly influenced the quality of the WWTP outflow being discharged into the Vaal River. Analysis of the results from the study indicate that the type of industry and activity undertaken therein influence the pH and elemental composition of the effluent. The seasonal variation of all parameters analysed for each sample location were not significant except for Zn and Al. It was also established in the study that the Leeuwkuil WWTP could not treat the inflow of wastewater received to acceptable quality, especially in lowering the concentration of Cu, Zn, Pb and BOD before discharging it into the Vaal River. Findings from this study indicate that heavy metal content in the industrial effluent contributed to the inefficiency of the WWTP. This is because (1) the current capacity of the WWTP was unable to handle the influent pollutant load and (2) the very low BOD/COD ratios of the influent—an indication of a high non-biodegradable content—could have influenced the biological effluent treatment processes. Adequate pre-treatment of effluents by industries before discharge would reduce the treatment burden and the cost involved as indicated in this study. A continuous monitoring of the diverse industrial effluents and river quality in the study area would help to identify the major contributing source of pollutants in the Vaal River. We therefore recommend continuous monitoring and maintenance of the WWTPs infrastructure in the study area. Further studies involving an analysis of heavy and trace metals in the sediment and pore-water sediment would help to understand the level of eco-toxicological risk inherent in the area. 

## Figures and Tables

**Figure 1 ijerph-17-01096-f001:**
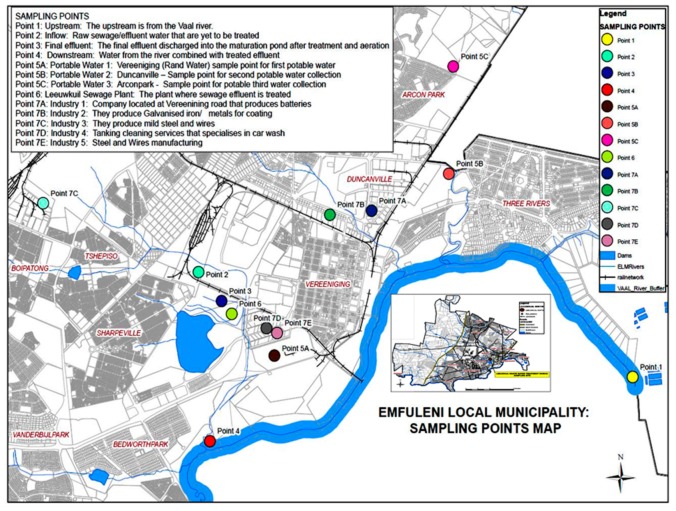
Map showing the study area and sampling points within Emfuleni Local Municipality, Gauteng Province, South Africa.

**Figure 2 ijerph-17-01096-f002:**
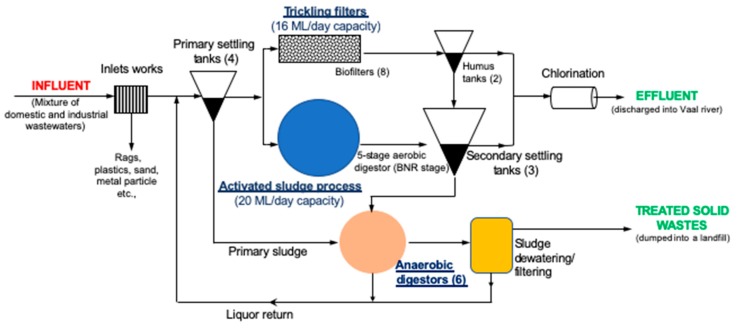
Flow diagram of the Leeuwkuil wastewater treatment plant (WWTP) showing stages of wastewater treatment, the capacity and the number of facilities within the plant (in brackets).

**Figure 3 ijerph-17-01096-f003:**
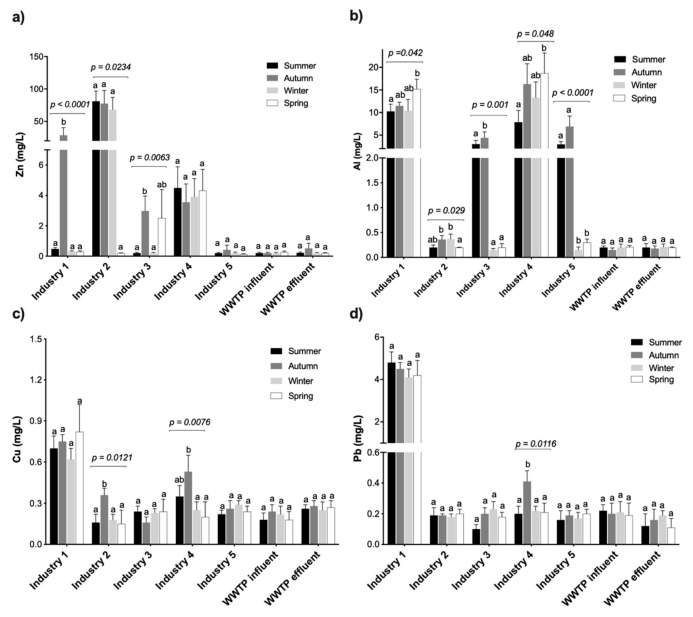
Seasonal variability of toxic heavy metal compositions in the industrial and Leeuwkuil WWTP discharges. Mean values followed by different letters (a and b) within a sampling point are statistically different (ANOVA; Tukey’s test, *p* < 0.05). *p*-Values of the waste streams showing significant differences between seasons are provided.

**Table 1 ijerph-17-01096-t001:** Summary of physicochemical characteristics (mg/L) in the different industrial wastewater and Leeuwkuil WWTP effluents of the Vaal triangle ^η^.

Sampling Site	Temp (°C)	pH	BOD (mg/L)	DO (mg/L)	COD (mg/L)	TDS (mg/L)	EC (mS/cm)	Salinity (psu)
Industry 1: Battery	22.2 ± 4.7 (16.4–27.4)	4.6 ± 0.8 ^a^ (3.1–6.2)	6.0 ± 0.6 ^a^ (5.6–6.8)	1.6 ± 0.3 ^a^ (1.2–1.9)	256 ± 62 ^a^ (184–320)	1868 ± 463 ^b^ (1301–2434)	258 ± 21 ^a^ (232–281)	3.0 ± 2.5 ^ab^ (1.4–6.8)
Industry 2: Iron/metal galvanizing	21.8 ± 4.6 (15.7–26.6)	8.0 ± 1.1 ^a^ (6.4–8.7)	6.0 ± 0.7 ^a^ (5.4–7.0)	1.7 ± 0.7 ^ab^ (0.7–2.1)	460 ± 476 ^abc^ (174–1172)	1827 ± 489 ^b^ (1138–2223)	2969 ± 834 ^ab^ (2223–3782)	1.9 ± 0.5 ^ab^ (1.2–2.4)
Industry 3: Iron/steel	22.3 ± 5.2 (16.2–28.2)	9.7 ± 0.5 ^ab^ (9.2–10.2)	5.4 ± 1.0 ^ac^ (4.2–6.6)	1.6 ± 0.5 ^a^ ((1.0–2.2)	1038 ± 304 ^c^ (872–1493)	16.8 ± 4.6 ^a^ (12–22)	30.8 ± 14.0 ^a^ (14–45)	20.2 ± 4.9 ^c^ (14.4–24.8)
Industry 4: Tanking/car wash	22.2 ± 5.1 (15.8–27.7)	10.9 ± 2.2 ^b^ (7.7–12.5)	5.9 ± 0.8 ^a^ (5.1–6.9)	2.0 ± 0.2 ^ab^ (1.8–2.1)	918 ± 294 ^bc^ (620–1209)	2330 ± 700 ^b^ (1417–3117)	2460 ± 1492 ^ab^ (1417–4673)	1.4 ± 0.4 ^ab^ (0.9–2.5)
Industry 5: Iron/steel	22.3 ± 5.1 (16.0–27.9)	11.1 ± 0.9 ^b^ (10.3–12.2)	5.9 ± 0.8 ^a^ (5.2–6.7)	1.2 ± 0.5 ^a^ (0.8–1.9)	310 ± 251 ^a^ (95–670)	3654 ± 891 ^c^ (2468–4611)	3849 ± 3756 ^b^ (1109–9224)	5.8 ± 0.5 ^b^ (5.2–6.2)
WWTP influent	21.6 ± 4.0 (16.6–26.3)	8.3 ± 0.6 ^a^ (7.8–9.0)	3.3 ± 1.2 ^b^ (2.1–4.9)	1.4 ± 0.3 ^a^ (1.0–1.7)	434 ± 50 ^ab^ (360–468)	335 ± 14 ^a^ (321–354)	590 ± 171 ^ab^ (337–710)	0.34 ± 0 ^a^ (0.33–0.35)
WWTP effluent	22.7 ± 5.0 (16.7–27.0)	8.0 ± 0.3 ^a^ (7.7–8.4)	3.8 ± 0.7 ^bc^ (3.1–4.4)	2.7 ± 0.4 ^b^ (2.2–3.0)	36 ± 67 ^a^ (0–136)	258 ± 21 ^a^ (232–281)	469 ± 143 ^ab^ (281–627)	0.26 ± 0.02 ^a^ (0.2–0.3)
*F-statistic* ^τ^	*0.02*	*8.06 ****	*7.32 ***	*5.05 ***	*7.61 ****	*28.63 ****	*3.88 ***	*44.38 ****
Green drop Standards ^#^	30	5.5–9.5	-	-	75	25	150	-
DWAF 2013 ^§^	-	5.5–9.5	-	-	75	25	70–150	-

^η^ Mean ± SD values followed by different letters with a column are statistically different (ANOVA; Tukey’s *t* test, *p* < 0.05). Range values is provided in bracket. ^τ^ Significance difference is denoted by *** *p* < 0.001, ** *p* < 0.01. ^#^ South Africa Department of Water and Sanitation (DWS) wastewater quality management limits to be achieved by Water Service Authorities and wastewater treatment facilities for certification [36]. ^§^ Legal permissible limits set by Department of Water Affairs and Forestry in South African Government Gazette No.36820, 2013 [27]. BOD, biological oxygen demand; DO, dissolved oxygen; COD, chemical oxygen demand; TDS, total dissolved solids; and EC, electrical conductivity.

**Table 2 ijerph-17-01096-t002:** Concentrations of major heavy metals (mg/L) in the different industrial wastewater and Leeuwkuil WWTP effluents of the Vaal triangle.

Sampling Asite	Al	Zn	Cu	Pb
Industry 1: Battery	11.8 ± 1.16 ^a^ (10.3–15.2)	7.41 ± 2.04 (0.5–28.5)	0.72 ± 0.04 ^a^ (0.62–0.81)	4.64 ± 0.17 ^a^ (4.3–5.0)
Industry 2: Iron/metal galvanizing	0.28 ± 0.05 ^b^ (0.2–0.37)	56.7 ± 19.03 ^a^ (0.2–81.1)	0.24 ± 0.04 (0.15–0.35)	0.18 ± 0.02 (0.15–0.23)
Industry 3: Iron/steel	1.95 ± 1.05 ^b^ (0.2–4.41)	1.47 ± 0.74 (0.2–2.97)	0.19 ± 0.02 (0.12–0.25)	0.18 ± 0.03 (0.1–0.26)
Industry 4: Tanking/car wash	14.0 ± 2.34 ^a^ (7.85–18.7)	4.07 ± 0.21 (3.55–4.49)	0.18 ± 0.02 (0.13–0.24)	0.19 ± 0.03 (0.13–0.25)
Industry 5: Iron/steel	2.56 ± 1.59 ^b^ (0.12–6.92)	0.26 ± 0.05 (0.20–0.41)	0.19 ± 0.02 (0.14–0.23)	0.21 ± 0.00 (0.2–0.21)
WWTP influent	0.18 ± 0.03 ^b^ (0.15–0.24)	0.20 ± 0.01 (0.18–0.23)	0.22 ± 0.02 (0.18–0.26)	0.20 ± 0.03 (0.12–0.23)
WWTP effluent	0.20 ± 0.02 ^b^ (0.15–0.24)	0.16 ± 0.03 (0.09–0.23)	0.18 ± 0.02 (0.14–0.24)	0.19 ± 0.03 (0.12–0.23)
*F–statistic* ^τ^	*23.42 ****	*7.31 ****	*46.58 ****	*584.45 ****
DWAF 2013 ^§^	-	0.1	0.01	0.01

^η^ Means followed by different letters (a, b) with columns are statistically different (ANOVA; Tukey’s *t* test, *p* < 0.05). ^τ^ Significance difference is denoted by *** *p* < 0.001. ^§^ Legal permissible limits set by South African Government Gazette No.36820, 2013 [36].

**Table 3 ijerph-17-01096-t003:** Two-tailed Pearson’s correlation coefficients for the physicochemical parameters and toxic heavy contents in the industrial and Leeuwkuil WWTP effluents ^§^.

Parameter	pH	BOD	COD	DO	TDS	EC	Salinity	Al	Zn	Cu	Pb
pH	1										
BOD	0.12	1									
DO	−0.02	**−0.47 ***	1								
COD	0.19	0.00	−0.06	1							
TDS	0.25	0.10	0.16	**−0.51 ***	1						
EC	0.41	0.04	0.13	−0.29	**0.66 ****	1					
Salinity	0.14	−0.31	−0.13	0.36	**−0.60 ****	−0.36	1				
Al	−0.03	0.25	0.17	0.10	0.08	−0.20	−0.35	1			
Zn	−0.24	0.10	−0.01	−0.05	0.05	0.22	−0.25	−0.32	1		
Cu	**−0.61 ****	0.09	−0.10	−0.35	−0.03	−0.37	−0.26	0.41	0.02	1	
Pb	**−0.57 ****	0.10	−0.04	−0.41	−0.02	−0.37	−0.23	**0.46 ***	−0.12	**0.96 *****	1

^§^ Correlations are defined as weak (0  <  |r|  <  0.3), moderate (0.3  <  |r|  <  0.7) or strong (|r |  >  0.7). Significant correlations (at *p* < 0.001 ‘***’, *p* < 0.01 ‘**’, *p* < 0.05 ‘*’) are bolded. BOD, biological oxygen demand; DO, dissolved oxygen; COD, chemical oxygen demand; TDS, total dissolved solids; and EC, electrical conductivity.

**Table 4 ijerph-17-01096-t004:** Two-tailed Pearson’s correlation coefficients for the physicochemical parameters and toxic heavy contents in the Leeuwkuil WWTP influent and effluent ^§^.

Parameter	pH	BOD	DO	COD	TDS	EC	Salinity	Al	Zn	Cu	Pb
pH	1										
BOD	0.43	1									
DO	−0.26	0.31	1								
COD	0.45	−0.19	**−0.95 *****	1							
TDS	0.35	−0.17	**−0.87 ****	**0.92 ****	1						
EC	−0.21	−0.05	−0.12	0.23	**0.66 ****	1					
Salinity	0.36	−0.22	**−0.90 ****	**−0.95 *****	**0.99 *****	0.31	1				
Al	−0.36	−0.03	0.53	−0.45	−0.14	0.20	−0.21	1			
Zn	−0.07	0.24	−0.55	0.44	0.39	−0.05	0.40	−0.29	1		
Cu	0.22	0.16	−0.67	0.56	0.35	0.00	0.40	**−0.88 ****	0.63	1	
Pb	0.56	−0.24	−0.11	0.20	0.08	−0.23	0.09	−0.37	0.64	0.01	1

^§^ Correlations are defined as weak (0 < |r| < 0.3), moderate (0.3 < |r| < 0.7) or strong (|r | > 0.7). Significant correlations (at *p* < 0.001 ‘***’, *p* < 0.01 ‘**’, *p* < 0.05 ‘*’) are bolded. BOD, biological oxygen demand; DO, dissolved oxygen; COD, chemical oxygen demand; TDS, total dissolved solids; and EC, electrical conductivity.

**Table 5 ijerph-17-01096-t005:** Physicochemical characteristics and mean concentrations of major heavy metals (mg/L) in Vaal River samples upstream and downstream of WWTP effluent discharge and potable water around the Vaal Area.

Parameter *	Upstream		Downstream		Potable Water 1		Potable Water 2		*p*-Value	SANS 241-1 2015 ^§^
	Mean ± SD	Range	Mean ± SD	Range	Mean ± SD	Range	Mean ± SD	Range		
pH	8.4 ± 0.45	7.8–8.8	8.5 ± 0.57	7.8–9.2	8.8 ± 0.36	8.5–9.2	8.9 ± 0.25	8.7–9.2	0.3521	5.0–9.7
BOD (mg/L)	4.497 ± 0.334 ^a^	4.1–4.9	4.305 ± 0.258 ^a^	4.–4.7	1.01 ± 0.142 ^b^	0.9–1.2	1.09 ± 0.553 ^b^	0.6–1.8	<0.0001	5
DO (mg/L)	3.548 ± 0.403 ^ab^	3.1–3.9	3.202 ± 0.708 ^a^	2.2–3.8	4.948 ± 0.618 ^c^	4.5–5.8	4.840 ± 0.810 ^bc^	4.1–6.0	0.0045	75
COD (mg/L)	88 ± 57.2 ^a^	83–155	162 ± 54.6 ^b^	105–230	7.75 ± 9.18 ^c^	0–18	10.28 ± 14.03 ^c^	0–31	0.0004	30
TDS (mg/L)	313 ± 38 ^a^	312–358	614 ± 174 ^b^	381–782	59 ± 24 ^c^	30–89	64 ± 25 ^c^	27–78	<0.0001	-
EC (mS/cm)	578 ± 160 ^a^	377–736	553 ± 130 ^a^	358–630	160 ± 78 ^b^	77–261	154 ± 80 ^b^	75–262	0.0002	250
Salinity (psu)	0.313 ± 0.033 ^a^	0.3–0.4	0.344 ± 0.033 ^a^	0.2–0.5	0.090 ± 0.027 ^b^	0.07–0.13	0.153 ± 0.127 ^b^	0.07–0.34	0.0005	-
Al (mg/L)	0.165 ± 0.044 ^b^	0.11–0.2	0.218 ± 0.092 ^ab^	0.14–0.35	0.078 ± 0.049 ^a^	0.012–0.13	0.060 ± 0.062 ^a^	0.05–0.15	0.0146	0.3
Zn (mg/L)	0.220 ± 0.091	0.12–0.34	0.380 ± 0.269	0.2–0.77	0.234 ± 0.047	0.2–0.3	0.310 ± 0.220	0.2–0.64	0.5541	5
Cu (mg/L)	0.119 ± 0.076 ^a^	0.03–0.21	0.166 ± 0.123 ^b^	0.01–0.28	<MDL ^α^		<MDL		0.0138	2
Pb (mg/L)	0.098 ± 0.033 ^a^	0.05-0.13	0.183 ± 0.041 ^b^	0.13-0.21	<MDL		<MDL		<0.0001	0.01

* Mean ± SD values followed by different letters (a, b, c) within a column are statistically different (ANOVA; Tukey’s HSD test, *p* < 0.05). ^α^ MDL, minimum detectable limit of the instrument. ^§^ South Africa national Standards for drinking/potable water [46].

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
