# Peer review of "Investigating Industrial Effluent Impact on Municipal Wastewater Treatment Plant in Vaal, South Africa"

_ijerph, 2020, doi:10.3390/ijerph17031096_

Round 1

Reviewer 1 Report

Although the study seems important to south Africa, there major flaws in the study.

A description of processes in the WWTP is missing. And so, if the WWTP is not well treating the metals contents of wastewater, so its obvious the correlation between the industrial discharges contents and river quality; it is not a conclusion.

What will more interesting, is not the correlation among same industry samples (although necessary), but the correlation within different sampling points. 

DO units are mg/l and not mg/ml (in 2 locations in the paper)

The major concern, is the study approaching the problem based on concentrations of different parameters. WWTPs' influent and effluent, and rivers quality depend on loads (mass/time) and not concentrations, low flows with high concentrations will results in medium loads, and so forth, also the time of discharge as industries do not discharge their wastewater continuously. It is advised for authors to address the issue based on the total maximum daily loads (TMDL), approach.  

Language of manuscript needs proof revision. 

Author Response

REVIEWER 1

Although the study seems important to south Africa, there major flaws in the study.

A description of processes in the WWTP is missing. And so, if the WWTP is not well treating the contents of the metal of wastewater, so its obvious the correlation between the industrial discharges contents and river quality; it is not a conclusion.

Authors response:

As suggested by the reviewer, we have added modified subsection 2.2 to include the information (Line 94-118) and a figure (Figure 2) on the capacity and stages of treatments including challenges faced by the plant in treating wastewaters ( Line 119-122)

Furthermore, the results and discussion relating to contribution of industrial wastewater vis-a-viz WWTP performance and their contribution to pollution of Vaal rive has been added (Line 221-225,278-281, 308-332)

What will more interesting, is not the correlation among same industry samples (although necessary), but the correlation within different sampling points.

 Authors response:

We have added Table 3. The correlation between WWTP influent and effluent to enrich results as suggested by reviewer (Line 357-366)

DO units are mg/l and not mg/ml (in 2 locations in the paper)

Authors response:

The DO units have been corrected as suggested by reviewer  ( line 210 and 213)

The major concern, is the study approaching the problem based on concentrations of different parameters. WWTPs' influent and effluent, and rivers quality depend on loads (mass/time) and not concentrations, low flows with high concentrations will results in medium loads, and so forth, also the time of discharge as industries do not discharge their wastewater continuously. It is advised for authors to address the issue based on the total maximum daily loads (TMDL), approach.

Authors response:

Generally, industrial water demand and wastewater production are sector-specific. Depending on the industrial process, the concentration and composition of the waste flows can vary significantly. Therefore, we agree with the reviewer that concentrations and flow/loads need to be taken in the count when interpreting results. We expanded discussion of the results to highlight this (Line 221-235, 308-332 ).

 Although part of the initial research objective to also collect data on the wastewater flow/data, many of the industries did not approve the collection of samples from the premises, and those that agreed did not approve collecting or using their wastewater discharge rate or load data into a municipal sewer line. However, it is estimated that about 10% of the hydraulic load into the WWTP plant is industrial wastewater. Since all industrial wastewaters, either pre-treated inhouse or not were discharged into the municipal sewer system, without the flow or load data, evaluation of the impact loading on WWTP treatment was not possible. We have included this as a limitation of the current study that needs further analysis if this data can be collected in future studies. The need to establish TMDLs given the limitations of the current study has also been highlighted (Line 421-432)

Language of manuscript needs proof revision.

Authors response:

The language was revised throughout the manuscript for better reading.

Reviewer 2 Report

I revised the paper titled: “Investigating Industrial Effluent Impact on Municipal Wastewater Treatment Plant in Vaal, South Africa”. The paper is interesting and has a good structure. Also, the discussion is good. In some points, English need a revision. However, I suggest major revision (expecially in the section of Materials and Methods) before a publication. My detailed comments are the following:

Line 17-20: Please rewrite this sentence. The English is not correct.

Line 20: Please, use “,” not “;” when there is a list.

Line 52-57: I suggest you some paper that they can helpful for you about contaminants in wastewater

https://doi.org/10.1007/s11270-019-4158-1

https://doi.org/10.1016/j.jhazmat.2019.121668

https://doi.org/10.1016/j.psep.2019.10.022

Line 112: What is Population Equivalent (P.E.) associated with the WWTP? What is the percentage of wastewater flow related with industrial origin? I also suggest inserting in section 2 some other info about the treatment present in the WWTP object of the study. Maybe, inserting a new figure could be useful.

Table 1: you write “Table 1. Summary of physicochemical characteristics and mean concentrations of major heavy metals (mg/l) in the different industrial wastewater” but mean concentration of major heavy metals are not presented. Please, amend it.

Please, identify in the table the type of activity of the industries 1, 2 , 3 and 4.

Figure 2: Please, insert in the caption the meaning of “a” and “b” written on the different column.

Author Response

REVIEWER 2

I revised the paper titled: “Investigating Industrial Effluent Impact on Municipal Wastewater Treatment Plant in Vaal, South Africa”. The paper is interesting and has a good structure. Also, the discussion is good. In some points, English need a revision. However, I suggest major revision (expecially in the section of Materials and Methods) before a publication. My detailed comments are the following: Line 17-20: Please rewrite this sentence. The English is not correct.

Authors response:

We have re-looked the draft and made necessary changes in relation to sentence structure and English as suggested by the reviewer (e.g. Line 15-19 among others).

Materials and methods have also been improved to include the information (Line 94-118) and a figure (Figure 2) on the capacity and stages of treatments including challenges faced by the plant in treating wastewaters ( Line 119-122)

Line 20: Please, use “,” not “;” when there is a list.

Authors response:

We have corrected as suggested in the abstract section where listed items had “;” instead “,”

Line 52-57: I suggest you some paper that they can helpful for you about contaminants in wastewater

https://doi.org/10.1007/s11270-019-4158-1

https://doi.org/10.1016/j.jhazmat.2019.121668

https://doi.org/10.1016/j.psep.2019.10.022

Authors response:

We have included additional relevant citations while removing non-relevant ones as suggested by all the reviewers.

Line 112: What is Population Equivalent (P.E.) associated with the WWTP? What is the percentage of wastewater flow related with industrial origin? I also suggest inserting in section 2 some other info about the treatment present in the WWTP object of the study. Maybe, inserting a new figure could be useful.

Authors response:

According to the feasibility studies done in 2017 to upgrade the Leeuwkuil WWTP, the built capacity is 36 000 m3/day ADWF; with an installed 20 000 and 16 000 m3/day of activated sludge plant and trickling filters, respectively [1] capable of handling 83 050 population equivalent. However, the plant receives approximately 42 000 m3/d average daily waste flow (ADWF), which is 116% beyond the design capacity. The estimated corresponding organic load of the plant is 15 262 kg COD/day, equivalent to the plant being organically overloaded by 153%. As suggested by the reviewer, we have added modified subsection 2.2 to include the information (Line 94-122) and a figure (Figure 2) on the capacity and stages of treatments including challenges faced by the plant in treating wastewaters. However, official information on hydraulic load contribution by industries to waste treated by the plant is lacking, as industrial wastes are mixed with domestic in the municipal sewers before reaching the treatment plant. However unofficial estimates are that they contribute to approximately between 0.89-10%.

GIBB The Sedibeng Regional Sanitation Scheme (SRSS): Leeuwkuil WWTW WML-Draft Scoping Report; Johannesburg, 2017.

Table 1: you write “Table 1. Summary of physicochemical characteristics and mean concentrations of major heavy metals (mg/l) in the different industrial wastewater” but mean concentration of major heavy metals are not presented. Please, amend it.

Authors response:

We have amended the Table 1 title on Line 204 by deleting “……and mean concentrations of major heavy metals…….” as suggested by the reviewer

Please, identify in the table the type of activity of the industries 1, 2 , 3 and 4.

Authors response:

In Figure 1 legend we had described industry 1, 2, 3, 4 and 5 as lead-acid battery company, galvanized iron/metal coating industry, steel and wire industry, tanking and car wash industry, and steel and wire manufacturing Company, respectively. For clarity, we have included this information in section 2.2 (Line128-132) industry type in Table 1 (Line 204-205) and Table 2 (line 260-261)

Figure 2: Please, insert in the caption the meaning of “a” and “b” written on the different column.

Authors response:

In Figure 2 caption Line 304-307, it is clearly is indicated that  “…..Mean values followed by different letters within a sampling point are statistically different (ANOVA; Tukey’s test, P < 0.05)……..”.  

Reviewer 3 Report

The work entitled “Investigating Industrial Effluent Impact on Municipal Wastewater Treatment Plant in Vaal, South Africa” presents the impact industrial wastewater with significant heavy metal concentration on municipal wastewater effluent.

The research objective suggested also in the title of this work its very interesting however, the research methodology and the findings are obvious, the authors should try to interpret in more detail the findings and to do additional research to highlight and prove the importance of such research.

I would like to list couple of comments and suggestions:

Measuring only concentration values is not really a strong proof, wastewater flows and loads in m3/d, kg/d should be taken into account also. For a research paper 74 references are to much the authors should use only relevant ones; The introduction part is to wide, describes to many aspects which are irrelevant for the research topic, only in the introduction part 47 references were used; The domestic wastewater treatment plant should be presented in more detail, P.E, treatment technology etc. 4. Heavy Metal Analysis -> literature citation is needed for method used In the methodology all the five industries should be named based on their activity, in the article Industry 1 as battery and some indication of metal industrial activities for Industries 2,3 and 5 are named. I think that the type of industry should be presented clearly thus the type of wastewater can be identified and characterized way better. The flows for industrial and domestic ww flows should be taken into account; Figure 2 y axis mgkg-1, why was that concentration used? The authors should list all the relevant causes that could affect the performance of a domestic WWTP when industrial WW is also present What technology should be applied to remove heavy metals form the WW etc…. The conclusion should also reflect listed in points the effects of industrial WW on the performance of a municipal wwtp, whch was the main objective of the study.

Taking into account the above listed points I would like to suggest for MAJOR Revision of the article.

Author Response

REVIEWER 3

The research objective suggested also in the title of this work its very interesting however, the research methodology and the findings are obvious, the authors should try to interpret in more detail the findings and to do additional research to highlight and prove the importance of such research.

Authors response:

As suggested by the reviewer, we have added modified subsection 2.2 to include the information (Line 94-118) and a figure (Figure 2) on the capacity and stages of treatments including challenges faced by the plant in treating wastewaters ( Line 119-122)

Furthermore, the results and discussion relating to the contribution of industrial wastewater vis-a-viz WWTP performance and their contribution to pollution of Vaal rive has been added taking into account pollutant concentrations and wastewater flow/loads (Line 221-224, 278-281, 308-332).

Measuring only concentration values is not really a strong proof, wastewater flows and loads in m3/d, kg/d should be taken into account also.

Authors response:

According to the feasibility studies done in 2017 to upgrade the Leeuwkuil WWTP, the built capacity is 36 000 m3/day ADWF; with an installed 20 000 and 16 000 m3/day of activated sludge plant and trickling filters, respectively [1] capable of handling 83 050 population equivalent. However, the plant receives approximately 42 000 m3/d average daily waste flow (ADWF), which is 116% beyond the design capacity. The estimated corresponding organic load of the plant is 15 262 kg COD/day, equivalent to the plant being organically over loaded by 153%. As suggested by the reviewer, we have added modified subsection 2.2 to include the information (Line 94-122) and a figure (Figure 2) on the capacity and stages of treatments including challenges faced by the plant in treating wastewaters. However, official  information on hydraulic load contribution by industries to waste treated by the plant are lacking, as industrial wastes are mixed with domestic in the municipal sewers before reaching the treatment plant. However unofficial estimates is that they contribute to approximately between 0.89-10%.

GIBB The Sedibeng Regional Sanitation Scheme (SRSS): Leeuwkuil WWTW WML-Draft Scoping Report; Johannesburg, 2017.

For a research paper 74 references are to much the authors should use only relevant ones; The introduction part is to wide, describes to many aspects which are irrelevant for the research topic, only in the introduction part 47 references were used;

Authors response:

As suggested by the reviewer, we have reworked the introduction section to be more focused to the study, shortening it considerably while citing only relevant literature. (line 35-85)

The domestic wastewater treatment plant should be presented in more detail, P.E, treatment technology, etc.

Authors response:

Materials and methods have also been improved to include the information (Line 94-118) and a figure (Figure 2) on the capacity and stages of treatments including challenges faced by the plant in treating wastewaters ( Line 119-122)

Heavy Metal Analysis -> literature citation is needed for method used In the methodology

Authors response:

The citation for the heavy metal analysis method has been included (line 147)

all the five industries should be named based on their activity, in the article Industry 1 as battery and some indication of metal industrial activities for Industries 2,3 and 5 are named. I think that the type of industry should be presented clearly thus the type of wastewater can be identified and characterized way better.

Authors response:

In Figure 1 legend we had provided the description of industry 1, 2, 3, 4 and 5 as lead-acid battery company, galvanized iron/metal coating industry, steel and wire industry, tanking and car wash industry, and steel and wire manufacturing Company, respectively. For clarity we have included this information in section 2.2 (Line128-132) industry type in Table 1 (Line 204-205) and Table 2 (line 260-261)

The flows for industrial and domestic ww flows should be taken into account;

Authors response:

Generally, industrial water demand and wastewater production are sector-specific. Depending on the industrial process, the concentration and composition of the waste flows can vary significantly. Therefore, we are in agreement with the reviewer that concentrations and flow/loads need to be taken in count when interpreting results. We expanded discussion of the results to highlight this (Line 221-235, 308-331 ).

 Although part of the initially research objective to also collect data on the wastewater flow/data, many of the industries did not approve collection of samples from the premises, and those that agreed did not approve collecting or using their wastewater discharge rate or load data into municipal sewer line. However, it is estimated that about 10% of hydraulic load into the WWTP plant is industrial wastewater. Since all industrial wastewaters, either pre-treated inhouse or not, were discharged into municipal sewer system, without the flow or load data, evaluation of the impact loading on WWTP treatment was not possible. We have included this as a limitation of the current study that need further analysis if this data can be collected in future studies. The need to establish TMDLs in view of the limitations of the current study has also been highlighted (Line 421-432)

Figure 2 y axis mgkg-1, why was that concentration used?

Authors response: In Figure 2  (now Fig 3) the y-axis concentration units was wrongly indicated mg.kg-1, since samples were liquid in nature, the concentration should have been correctly denoted as mg/L. We have corrected this accordingly in the Figure 3 (line 3)

The authors should list all the relevant causes that could affect the performance of a domestic WWTP when industrial WW is also present. What technology should be applied to remove heavy metals form the WW etc….

Authors response:

We have added a discussion on relevant causes that could affect the performance of a domestic WWTP when industrial WW is also present  and tertiary treatment techniques for heavy metal as suggested (line 308-331).

The conclusion should also reflect listed in points the effects of industrial WW on the performance of a municipal wwtp, whch was the main objective of the study.

Authors response

We have added two major points based on findings from the study in the conclusion section. The points included are: the current capacity of the WWTP was unable to handle the influent pollutant load arising from the high heavy metal content from the influent and (ii) the very low BOD/COD ratios of the influent -an indication of a high non-biodegradable content- could have influenced the biological effluent treatment processes.

Round 2

Reviewer 1 Report

Overall, the manuscript is majorly enhanced. 

Line 174, "thereby may result to the reduction of the efficiency of nutrient removal", revise English.

Please reconsider location of figures 1 and 2, t be after the paragraph where they are mentioned.

Need to confirm in the paper that treated sludge (solids) by anaerobic digesters, or there liquors  are not discharged to Vaal river, otherwise this will be a major source of heavy metals contamination in the river and need to be quantified and discussed in the paper

In line 348, add "(Pearson)" after "-tailed" 

Please add reference on  EPA-TMDL; line 426.

Author Response

REVIEWER 1

Line 174, "thereby may result to the reduction of the efficiency of nutrient removal", revise English.

Author response: The sentence have been modified to read   “……….during the treatment process, that may reduce waste treatment efficiency”. Line 171-173

Please reconsider location of figures 1 and 2, t be after the paragraph where they are mentioned.

Authors response: We have modified the location of figure 1 (line 95) and 2  (line 107)as suggested by the reviewer

Need to confirm in the paper that treated sludge (solids) by anaerobic digesters, or there liquors  are not discharged to Vaal river, otherwise this will be a major source of heavy metals contamination in the river and need to be quantified and discussed in the paper

Authors response: In Leeuwkuil WWTP, upon anaerobic digestion of sludge the liquor/supernatant i recycled to the primary settling tank  for retreatment. In addition, liquor from sludge dewatering plant is also retreated. The treated solid sludge wastes are also transported and dumped into a landfill, therefore the contribution of the sludge to the heavy metal contamination of the Vaal river is minimal. We have modified Figure 2, to illustrate the handling of anaerobic process and sludge dewatering plant liquor (Line 106).

In line 348, add "(Pearson)" after "-tailed" 

Authors response: We have corrected line 160, 341, 347 and 362 as suggested by the reviewer

Please add reference on  EPA-TMDL; line 426.

Authors response: We have added two references on TMDLs in line 428 as suggested by the reviewer

Reviewer 2 Report

The authors revised the manuscript in accordance with my comments. Considering the improvments, the manuscript can be published.

Author Response

We thank the reviewer for supporting our work.

Reviewer 3 Report

The authors work entitled “Investigating Industrial Effluent Impact on Municipal Wastewater Treatment Plant in Vaal, South Africa” was updated taking into account my comments and suggestions. The authors responded to all my comments, with double check of English language and grammar I propose to publish the paper, however I strongly advice to the authors to continue the research and use the current findings as a start of a more detailed research (process engineering, environmental monitoring etc. studies).

Author Response

REVIEWER 3

The authors work entitled “Investigating Industrial Effluent Impact on Municipal Wastewater Treatment Plant in Vaal, South Africa” was updated taking into account my comments and suggestions. The authors responded to all my comments, with double check of English language and grammar I propose to publish the paper, however I strongly advice to the authors to continue the research and use the current findings as a start of a more detailed research (process engineering, environmental monitoring etc. studies).

Author response: We are in agreement with the reviewer that further in-depth studies on WWTP process engineering and environmental monitoring are needed to build on the data collected in this study. The research team and their collaborators are in the process of sourcing funding to cover the different areas suggested.